# Genetic and Epigenetic Mechanisms of Psoriasis

**DOI:** 10.3390/genes14081619

**Published:** 2023-08-13

**Authors:** Laura Mateu-Arrom, Lluis Puig

**Affiliations:** Department of Dermatology, Hospital de la Santa Creu i Sant Pau, Institut d’Investigació Biomèdica Sant Pau (IIB SANT PAU), Universitat Autònoma de Barcelona, 08041 Barcelona, Spain

**Keywords:** psoriasis, genetics, epigenomics

## Abstract

Psoriasis is a disease involving the innate and adaptative components of the immune system, and it is triggered by environmental factors in genetically susceptible individuals. However, its physiopathology is not fully understood yet. Recent technological advances, especially in genome and epigenome-wide studies, have provided a better understanding of the genetic and epigenetic mechanisms to determine the physiopathology of psoriasis and facilitate the development of new drugs. This review intends to summarize the current evidence on genetic and epigenetic mechanisms of psoriasis.

## 1. Introduction

Psoriasis is a chronic inflammatory skin disease affecting from 0.11 to 1.58% of individuals worldwide, depending on the region [1]. Although multiple types of psoriasis have been reported, including plaque psoriasis (psoriasis vulgaris), guttate, inverse, pustular, and erythrodermic psoriasis [2], scientific research refers predominantly to the psoriasis vulgaris variant, which represents approximately 85 to 90% of all psoriatic patients [3]. Typically, it appears as elevated, clearly defined, erythematous oval plaques with silvery scales firmly attached resulting from abnormal keratinocyte hyperproliferation and differentiation, accompanied by activation and infiltration of immune cells [4].

Psoriasis is associated with a significant psychological burden. In fact, health-related quality of life is impaired more in psoriasis than in other chronic diseases such as neoplasms or cardiopathy [2]. In a recent online survey conducted on almost 5000 patients with moderate to severe psoriasis, nearly half of all patients stated that psoriasis had a very large to extremely large impact on their quality of life [5].

Furthermore, psoriasis has been related to several comorbidities such as psoriatic arthritis, hypertension, diabetes mellitus or cardiovascular diseases [6,7], asthma, chronic obstructive pulmonary disease [8], among others. Thus, psoriasis carries a great burden on society and on health care systems [3].

The etiopathogenesis of psoriasis is complex and not completely understood. Psoriasis is nowadays considered an immune-mediated disease where a complex interaction between the innate and adaptive immune system arises from the interplay of inherent genetic predisposition and environmental risk factors [6,9].

Recently, significant progress has been achieved in uncovering the molecular mechanisms of psoriasis. Genome-wide studies have brought to light new genes associated with psoriasis and have demonstrated the involvement of multiple genes in the pathogenesis of the disease, with therapeutic implications [10]. At the same time, recent technological advances have enabled a more in-depth and sensitive exploration of the epigenetic mechanisms that regulate psoriasis gene expression at transcriptional and post-transcriptional stages [11]. This greater genetic and epigenetic understanding offers substantial potential for the identification of new therapeutic targets for psoriasis.

The aim of this review is to summarize the current evidence on the genetic and epigenetic mechanisms of psoriasis.

## 2. Material and Methods

An electronic literature search was conducted on the Medline/PubMed database until June 2023 using Medical Subject Headings terms and relevant medical terminology. The search criteria included the terms ‘psoriasis’, ‘genetic testing’, ‘genomics’, ‘genetics’, ‘epigenomics’, ‘epigenetics’. We considered original genome studies, reviews, systematic reviews and meta-analyses specifically related to the genetic or epigenetic study of psoriasis. English language manuscripts were included. On the other hand, letters to the editor, editorials, experts’ opinions, congress proceedings, studies including patients with psoriatic arthritis alone, those related to specific psoriasis treatments or focused on transcriptomics, or proteomics analysis were excluded. The selection of publications was performed by two independent researchers (LM, LP) and discrepancies were resolved by consensus.

A total of 321 articles were initially identified, from which 18 were duplicates/redundant and therefore excluded. Studies that did not offer pertinent information according to the research objectives (163) were excluded, as were 34 studies focused on psoriatic arthritis. After a comprehensive review of the full-text articles, 84 studies were finally eligible for inclusion in the review.

## 3. Results

### 3.1. Genetics

Psoriasis is significantly more likely to occur in first- and second-degree relatives of patients with psoriasis than in the general population [9,12], and concordance is greater in monozygotic than in dizygotic twins [9,13], pointing to a strong genetic influence on the disease. In fact, the heritability of psoriasis has been estimated to be higher than 60% [9,13], even though psoriasis is not a monogenic disease, but a complex and multifactorial disease involving multiple susceptibility genetic loci. Early analyses on this topic were carried out by family-based linkage disequilibrium studies [14], since genetic variants or loci which are in situated closely on the same chromosome are less prone to separation by recombination during meiosis, thus they are more likely to be inherited together and exhibit correlation in the population [9]. Areas thought to harbor psoriasis-related genes with psoriasis susceptibility were primarily named *PSORS* (psoriasis-susceptibility) loci [15]. There are at least 12 distinct *PSORS* loci that were mainly identified through linkage analysis of multiply affected psoriasis families [15]. Loci in the *major histocompatibility complex* (*MHC*) I, on the short arm of chromosome 6, were among the first and most repetitive genetic susceptibility regions found in psoriasis [10]. In fact, the first loci linked to psoriasis susceptibility, *PSORS 1*, was the human leukocyte antigen (*HLA*)-*Cw6*, situated at chromosomal position 6p21.3 [15,16].

Following the completion of the Human Genome Project in 2001, advancements in technology have facilitated the affordable study of the entire genome instead of isolated variants or single genes [17]. Genome-wide association studies (GWAS) analyzed several million genetic markers or single nucleotide polymorphisms (SNP) across the genome in large populations, allowing for the detection of subtle differences in allele frequencies between psoriatic individuals and healthy controls [9]. GWAS allowed for the confirmation of loci previously identified by linkage analysis and identified new loci related to psoriasis susceptibility, beyond the *MHC* [18,19]. These findings have significantly propelled the comprehension of disease mechanisms and related pathways. Specifically, apart from the MHC system, the genes found to be related with psoriasis belong to the pathways of skin barrier, innate immune system with particular relevance of nuclear factor-kappa B (NF-kB) and interferon (IFN) signaling, and the adaptative immune system, with an implication of CD8+ and CD4+ T cells, especially Th17 [10]. Table 1 summarizes the main genes associated with psoriasis.

#### 3.1.1. Antigen Presentation

The initial noteworthy genetic association of psoriasis was identified in the *MHC I* region [10]. MHC I molecules are present on almost all nucleated cells and their function is to present intracellular self or non-self peptides to the immune system [15,20]. This mechanism allows MHC-I to present numerous peptides, collectively termed the ‘immunopeptidome’. CD8+ T lymphocytes interpret the immunopeptidome by attaching to the peptide-MHC-I complexes using their T cell receptors (TCR) [20].

The *MHC* region accounts for a third of the total genetic influence on psoriasis [10], highlighting the role of the antigen presentation pathway in the pathogenesis of psoriasis. Within the *MHC*, the allele *HLA-C*0602* exhibits the most robust correlation with psoriasis [21] and is considered the *PSORS1* risk variant [10]. The *HLA- C*0602* allele is found in 20–50% of patients with psoriasis, varying according to the population studied, and in only up to 16% of controls [15]. This allele has also been related to guttate psoriasis, Koebner phenomenon, psoriasis improvement during pregnancy, psoriasis nail disease, and poor response to treatment [10]. In fact, the strong genetic linkage of psoriasis with *MHC-I* alleles provides the basis for considering this disease among the MHC-I-opathies, a group of conditions distinguished by disease-linked *MHC-I* alleles that present distinct self-immunogenic peptides leading to the initiation of autoimmune responses [20]. Indeed, CD8+ T cells that react against melanocyte-derived antigens have been identified in psoriasis associated with *HLA-C*0602* [20,22,23]. Other pathogenic theories suggest that MHC-I molecules could misfold and accumulate in the endoplasmic reticulum or that the predisposing MHC-I proteins could be recognized by killer-cell immunoglobulin-like receptors (KIRs) or leucocyte immunoglobulin-like receptors (LILRs) on the cell surface of natural killer (NK) cells [20].

Other loci related to antigen presentation have been associated with psoriasis susceptibility. GWAS have unveiled non-additive gene-to-gene interactions between the *HLA-class I* risk alleles and specific variations of the *endoplasmic reticulum aminopeptidase 1* (*ERAP1*), which can be linked with psoriasis risk [22]. *ERAP1* encodes an aminopeptidase that participates in N-terminal trimming of MHC-I-binding peptides in the endoplasmic reticulum to prepare them for effective antigen presentation [21,22].

These associations imply that psoriasis could be caused by a T-cell initiated response to an autoantigen that is preferentially exhibited on HLA-C*0602 and processed by specific *ERAP1* allele transcripts [15]. However, there is not a unique confirmed “auto-antigen” [15]. On the other hand, the T-cell infiltrate in psoriasis plaques is extensively polyclonal and lacks a predominant clonal growth of any specific T-cell responding to a distinct epitope [15]

Other loci found in the proximity of the *MHC* region are associated with an increased risk of psoriasis [24]. The *MHC class I polypeptide-related sequence A* (*MICA*) is found near the *HLA-B* locus in the *MHC* region. Its expression is believed to be triggered by stress, and it is a ligand for NKG2D, an activating receptor present on natural killer cells, NKT-cells, and T-cells [15].

#### 3.1.2. Th1 Signaling Pathway

Psoriasis has traditionally been considered a Th1 mediated disease [10], given that there is an elevation of T-cells producing interferon (IFN)-γ in psoriatic lesions [25]. Myeloid dendritic cells, stimulated by factors such as tumor necrosis factor (TNF)-α, IFN-α, IFN-λ, interleukin (IL)-6, and IL-1β instruct T-cells to differentiate to Th1 cells and secrete IL-12 [10].

Several genetic psoriasis susceptibility loci involved in different steps of the Th1 signaling pathway have been identified, including *IL12B* [26] and *TYK2* [27] (coding for the p40 subunit of IL-12 and tyrosine kinase 2, involved in signal transduction following activation of the IL-12 and other receptors), *ZC3H12C* [28] (transcript involved in macrophage activation), *STAT5A* and *STAT5B* (coding for the corresponding signal transducer and activator of transcription molecules involved in signaling of IL-2 family cytokines such as IL-2, IL-7, IL-15, and IL-21), and *ILF3* (coding for a nuclear transcription factor determining IL-2 expression in T cells) (Table 1) [10].

#### 3.1.3. Th17 Signaling Pathway

The discovery of Th17 and the Th17 polarizing cytokine IL-23 brought to light the crucial role of the IL-23/IL-17 pathway in psoriasis [25,29]. Transforming growth factor (TGF)-β, IL-6, IL-1β, and IL-23 promote Th17 cell differentiation and production of IL-17, which has pro-inflammatory effects mediated by activation of dendritic cells, macrophages, endothelial cells, chondrocytes, fibroblasts, and osteoblasts [10]. Both IL-17 and IL-23 interact with the innate immune system by activating nuclear factor kappa B (NF-κB) transcription factor [30].

The IL-23 cytokine consists of subunits p19 and p40, whereas the IL-23 receptor is a heterodimer composed of IL-12RB1 (shared with the IL-12) and IL-23R [15]. Several SNPs associated with psoriasis risk have been found in genomic regions corresponding to both subunits of IL-23 (p19 and p40) and IL23R [26,31]. It is worth mentioning that IL-12 consists of two subunits, p35 and p40, the latter is shared with IL-23 and encoded by the *IL12B* gene; initial investigations into IL-12 function, employing either anti–IL-12p40 antibodies or IL-12p40-deficient mice, assigned roles to IL-12 that are now accurately attributed to IL-23 [31].

The IL-23 receptor lacks inherent signaling capability and relies on interactions with downstream molecules (Tyk2 for IL-12RB1 and Jak2 for IL-23R). Signaling results in phosphorylation, activation, and nuclear translocation of STAT3 [15]. Mutations in genes coding for Tyk2, Jak2, and STAT3 have been found to be associated with psoriasis [32,33]. Other genetic variants associated with psoriasis and related to IL-23 correspond to *SOCS1* (coding for a suppressor of cytokine signal involved in Th17 differentiation) and *ETS1* (coding ETS protooncogene 1 protein, involved in the negative regulation of Th17 differentiation) [10].

To date, an association of the *IL17A* gene with psoriasis has not yet been established. However, a SNP in the *IL17R* is associated with disease susceptibility [34]. IL-22 is a Th17 cytokine that leads to differentiation and proliferation of keratinocytes and has been found to be upregulated in psoriatic skin [35,36]. *IL22* has been associated with psoriasis by GWAS, with a quantitative effect: the greater the copy number of the *IL-22* gene, the greater the risk of nail disease [32].

Neutrophils have been shown to play a role in the pathogenesis of psoriasis through different mechanisms including the generation of reactive oxygen species, degranulation, and the formation of neutrophil extracellular traps (NETs) [37]. The formation of NETs consists of the neutrophil death and extrusion of decondensed chromatin with histones and granule proteins forming a web-like structure that traps microorganisms and accumulates inflammatory mediators [37,38]. A psoriasis risk variant in the *TRAF3 interacting protein 2 (TRAF3IP2)* gene has been shown to induce Th17 differentiation and IL17A secretion, which is potentiated in the presence of NETs [39]. Other loci related to the IL-17/IL-23 axis and involved in psoriasis susceptibility correspond to genes coding for IL-6, with a protective effect, or *KLF4* genes [10].

In psoriasis patients there is an altered balance of regulatory T cells (Tregs) and Th17 cells [40,41,42,43]. Tregs are lymphocytes that regulate the immune system and suppress the immune response [40], thus preventing development of autoimmune disease. Tregs inhibit the activity of CD4+ and CD8+ T cells leading to decreased production of pro-inflammatory cytokines [28]. Some polymorphisms identified in genes such as *TNF, IL12RB2*, and *IL12B* could be also involved in the production, differentiation, or activity of Tregs, pointing to a genetic background of Treg dysfunction in psoriasis.

#### 3.1.4. Innate Immunity

The initial stimulation of the innate immune system is necessary for the subsequent activation of the adaptative system [15]. Many SNPs have identified gene candidates in psoriasis corresponding to innate immunity pathways. Transcription of these innate immune genes in psoriatic patients could lower the threshold required to trigger the pathogenic adaptive immune response [15].

NF-kB mediates the transcription of numerous genes involved in intracellular signaling and is one of the most important factors related to the innate response. The NF-kB pathway is activated in psoriatic lesions and numerous genes that encode elements of the NF-kB pathway are linked to psoriasis [44,45]. Among the genes associated with the activation of the NF-kB pathway, *C-REL* stands out and is also related with keratinocyte growth [46], and *TRAF3IP2* [47], highlighting the interactions between innate and adaptative immune system, or genes encoding CARD proteins [48].

Some genes associated with psoriasis susceptibility are related not with activation of the NF-kB pathway but with its downregulation; mutations in negative regulators resulting in reduced ability to control inflammation may be as important as mutations resulting in overactive immune responses [11]. The former genes include those coding for TNF-α-inducible protein 3 (TNFAIP3) and TNFAIP3 interacting protein 1 (TNIP1), which prevent the degradation of the NF-kB inhibitor [49]; *the NF-kB inhibitor α* (*NFKBIA*) which encodes the inhibitor of NF-kB signaling [50]; and the *zinc finger DHHC-type containing 23* (*ZC3H12C*), coding a protein that inhibits vascular inflammation [28].

The associations of innate immune genes with psoriasis have also been identified within the interferon pathway and antiviral response genes, such as *IFIHI* and *DDX58*. They encode innate pattern-recognition receptors, namely RIG-I and MDA5, which recognize viral RNA with subsequent activation of an anti-viral response, ending up in induction of type-I interferons and other innate inflammatory molecules [51]. These mutations could potentially lower the threshold for initiating responses to self- or non-self stimuli, potentially serving as a critical initial stage in lesion development [15].

#### 3.1.5. Skin Barrier Function

The role of the immune system is crucial in the pathogenesis of psoriasis, but there is a component of epithelial disorder, since epidermal keratinocytes are strongly implicated in the onset of psoriasis [10,21]. Polymorphisms in skin barrier regulatory genes such as defensins, late cornified envelope (LCE) genes, and connexin have been identified. The *DEFB4* gene is overexpressed in the psoriatic skin leading to an increase in secretion of β-defensins in response to Th1 or Th17-related mediators, and variations in *DEFB4* gene copy numbers are associated with psoriasis risk [52]. *LCE* genes, specifically *LCE3B* and *LCE3C*, located in the *PSORS4* locus, have been shown to be related with psoriasis risk by both copy number variation studies and GWAS [21,53]. Finally, a variant in the *GJB2* gene, encoding a connexin, was found as a psoriasis risk locus by GWAS [49].

#### 3.1.6. Genetics of Generalized Pustular Psoriasis

In contrast to plaque psoriasis, the pathogenesis of pustular psoriasis seems to be closely related to an autoinflammatory disorder with special contribution of neutrophils and participation of IL-36 cytokines, which in turn explains the differences in their clinical presentation [54]. The mutations underlying the presentation of generalized pustular psoriasis as a monogenic autoinflammatory disorder are also specific. Genetic variants related to generalized pustular psoriasis susceptibility include *IL36RN*, resulting in DITRA (Deficiency of The Interleukin-36-Receptor Antagonist) a monogenic autoinflammatory disorder [54] linked with a more severe manifestation of the disease and earlier onset age [21], and *CARD 14* (caspase recruitment domain family member 14, which is primarily expressed in keratinocytes and activates NF-kB signaling. Mutations activating CARD 14 have also been found in plaque psoriasis, psoriatic arthritis, and localized pustular psoriasis, besides generalized pustular psoriasis [54]. Mutations in the *adaptor-related protein complex 1 subunit sigma 3* (*AP1S3*) have also been found in generalized pustular psoriasis. Keratinocytes deficient in *AP1S3* may exhibit disrupted autophagy, resulting in significant elevation of the IL-1 signaling pathway and IL-36 cytokines [55]. Myeloperoxidase (MPO), a lysosomal hemoprotein present in neutrophilic granules, is involved in antimicrobial activity [54]. Mutations in *MPO* genes causing MPO deficiency, found in cases of pustular psoriasis [56], may lead to increased activity of neutrophilic proteases [57]. Other genes involved in pustular psoriasis are *SERPINA1* and *SERPINA3* (serine protease inhibitors that inhibit cathepsin G, neutrophil elastase, and proteinase 3), *TNIP1* and *IL1RN* [54].

#### 3.1.7. Genetics in Psoriatic Arthritis

Numerous genetic risk loci associated with innate, adaptive immune, and autoinflammatory pathways are shared between psoriasis and psoriatic arthritis [58,59]. However, the genetic variants specifically related to psoriatic arthritis have also been reported, and patterns of cytokine gene expression have been demonstrated to vary between skin and synovial tissue [58]. Whereas the correlation *of HLA-C*06:02* with psoriasis is more pronounced than with psoriatic arthritis, other HLA variants such as *HLA-B*27, HLA-B*39, HLA-B*38*, and *HLA-B*08* have been related with the development of psoriatic arthritis [58]. Moreover, the *HLA-B*08.01* allele has been linked to asymmetric sacroiliitis, peripheral arthritis, ankylosis and increased join damage, and *HLA-B*27* with symmetric sacroiliitis, dactylitis, and enthesitis [60]. On the other hand, polymorphisms in *IL23R, TNFAIP3, PTPN22* (*tyrosine-protein phosphatase non-receptor type 22), IL12B, RUNX1* (CD8-lymphocyte activation and differentiation), *IL13, KIR* (killer-cell immunoglobulin-like receptor), and *MAGI1* genes have shown association with psoriatic arthritis [10,58]. Also, variants in the *ADAMTS9* gene, involved in the cartilage extracellular matrix, are found to be associated with psoriatic arthritis [58].

#### 3.1.8. Genetics and Psoriasis Treatment

The substantial wealth of knowledge obtained through genetic research has provided essential insights into the biology of psoriasis, allowing for the development of increasingly effective therapeutic agents for psoriasis [61]. The genetic polymorphisms responsible for treatment response to the different available psoriasis treatment options have been identified [62,63,64], including those related to the risk of developing antidrug antibodies [65], and have been found to correspond in many cases with type I HLA molecules. At the same time, genetic susceptibility to adverse drug reactions has also been described [64,66]. Since a wide range of therapeutic alternatives are available for psoriatic disease, pharmacogenomic tests prior to treatment might facilitate selection of the therapy based on individual probabilities of drug response or adverse events, providing opportunities for personalized medicine.

### 3.2. Epigenetics

Epigenetics involves the study of reversible and heritable modifications in gene expression that do not stem from alterations in the DNA sequence [11]. Epigenetic mechanisms induce gene expression changes at a transcriptional or post-transcriptional level through modifications of DNA and histones, but not altering the DNA sequence, which change chromatin structure and activate transcription factors [67]. Epigenetic mechanisms are sensitive to external stimuli and represent the link between genetics and the environment [68]. In fact, in psoriasis, where not only genetics but also environmental factors play etiopathogenetic roles, epigenetics can modulate individual gene expressions thus modifying the likelihood of the disease [67]. This is especially important during embryogenesis when the level of pluripotency is elevated. However, as cell differentiation progresses, the epigenome stabilizes and becomes less susceptible to environmental conditions [69]. The role of epigenetics in psoriasis is highlighted by the low concordance (35–72%) between monozygotic twins, who share an identical DNA sequence [70,71,72]. Epigenetic mechanisms in psoriasis can interfere with gene transcription, including DNA methylation and histone modifications, or translation, through non-coding RNAs that include microRNA (miRNA), long non-coding RNA (lncRNA), and circular RNA (circRNA) [67]. Figure 1 summarizes the epigenetic mechanisms of DNA methylation and histone modification.

#### 3.2.1. Non Coding RNA

Non-coding RNAs (ncRNAs) are RNAs that do not undergo translation into proteins; instead, they function by interacting with RNA, DNA and proteins, inducing changes in their structure and impacting gene expression [4]. Depending on size, ncRNAs can be classified into short ncRNAs, long ncRNAs (lncRNAs), and intermediate ncRNAs. Table 2 summarizes main ncRNAs found in psoriasis.

Short ncRNAs include short interfering RNAs (siRNAs), Piwi-interacting RNAs (piRNAs), and microRNAs (miRNAs); the latter are the most relevant regarding psoriasis. Micro RNAs are 17–25 nucleotides long and regulate gene expression, mainly gene silencing, by binding to and eventually leading to degradation of mRNA, modulating methylation of DNA or targeting enzymes necessary for DNA methylation, or histone modifications [4,73].

miRNA125b, one of the most downregulated miRNAs in psoriatic lesions, has been associated with suppression of keratinocyte proliferation and promotion of keratinocyte differentiation [74,75]. However, many other dysregulated miRNAs, have been shown to be involved in psoriasis pathogenesis, modulating the immune response and skin inflammation as well as epithelial differentiation and proliferation [76]. Among them, miR-230 [77], mir-383 [78], Mir-214-3p [79], or miR-125a-5p [80] are downregulated; whereas, miR-378a [81], Mir-31 [82], mir-210 [83], miR-200c [84], or miR-155 [85] are upregulated in psoriasis.

Long non-coding RNAs are longer than 200 nucleotides and modulate gene expression by attracting transcription factors and proteins that modify chromatin structure [4] or acting as competitive endogenous RNAs constraining the accessibility of miRNAs to suppress their target genes [73]. Differentially expressed lncRNAs in psoriasis play immunological and epidermal differentiation functions [4]. lncRNA-RP6- 65G23.1 is a significant upregulated lncRNA in psoriasis that promotes keratinocyte proliferation and suppression of apoptosis by modifying the expression of Bcl-xl, Bcl2 and the ERK1/2-AKT signaling pathway [86]. Other upregulated lncRNA in psoriasis are MIR31HG [87], MSX2P1 [88], XIST [89], FABP5P3 [90], KLDHC7B-DT [91], or SPRR2C [92]; whereas, MEG3 [93], GAS5 [94], PRINS [95] or NEAT1 [96] are downregulated in psoriasis.

Circular RNAs (circRNAs) are intermediate single-stranded ncRNAs characterized by their covalent linkage between the 5′ and 3′ ends, forming a continuous circular structure [73], They facilitate the elimination of miRNAs or RNA-binding proteins, thus modulating gene expression or the translation of regulatory proteins [97]. circRNAs can also engage with proteins such as RNA polymerase II to facilitate transcriptional regulation [97]. Dysregulation of circRNAs in psoriasis has been associated with skin inflammation, keratinocyte hyperproliferation, and disease severity [73]. Some circRNAs, such as circRAB3B, which prevents keratinocyte hyper-proliferation through the upregulation of the tumor suppressor gene PTEN, are less abundant in psoriatic skin compared to non-lesional and healthy skin [98]. Conversely, circOAS3 [99], circEIF5 [100], circ_0060531 [101], hsa_circ_0003738 [102], or hsa_circ_0061012 [103] are upregulated in psoriasis.

#### 3.2.2. DNA Methylation

DNA methylation consists of the addition of a methyl group to the cytosine 5′ position of CpG dinucleotides through the action of DNA methyltransferases (DNMTs) [104]. This process allows for the binding of methylcytosine-binding proteins (MBPs), leading to chromatin compaction [104]. Methylation occurring in the promoter region of a gene results in reduced binding of transcription factors responsible for enhancing transcriptional activity, and subsequent repression of gene transcription [104]. Conversely, loss of DNA methylation in this region leads to re-activation of gene expression [104].

DNA methylation in psoriasis has been shown to occur in multiple immune active locations besides the skin and has been linked to disease severity, distribution of lesions and different tissue types implicated [105]. Epigenome-wide methylation studies have discovered differentially methylated genes and pathways in psoriatic lesions compared with healthy controls or psoriatic lesions compared with the non-lesional skin of psoriatic patients [69,106,107,108]. However, compared with psoriatic lesions, the methylation pattern of non-lesional skin of psoriatic patients was more similar to that of healthy patients [105].

When comparing psoriasis-involved skin to the adjacent normal skin of patients with psoriasis, differentially methylated CpGs are found, some of which situated at the promoters of known PSORS genes, such as *S100A9, PTPN22, SELENBP1, CARD14*, and *KAZN* [108]. Characteristically, psoriatic skin with Munro’s microabscess has shown to be abundant in differentially methylated genes related to neutrophil chemotaxis, suggesting that the DNA methylation profile can also be related with psoriasis histopathological features [108]. Other differentially methylated loci in psoriatic lesions compared with the normal skin of patients with psoriasis and with normal skin from healthy controls are *S100A8* (involved in epidermal differentiation), *CYP2S1* (metabolism of retinoic acid), and *EIF2C2* (with a role in RNA processing) [107].

Differential methylation patterns can also be found in psoriatic scales, as hypomethylation of CpG sites in psoriatic scales when compared to psoriatic skin lesions is observed, which is closely related to disease severity [109]. Such is the case of the *MGRN1* gene, which codes a protein involved in degradation of misfolded proteins [109].

It has been suggested that resolved psoriasis lesions generate local disease memory, involving the development of tissue-resident memory T cells in the skin, and prompting local relapse. Since the DNA methylation pattern does not change completely in lesional skin after treatment, epigenetic changes could contribute to this local memory [110]. In fact, differential methylation patterns have been identified between paired never-lesional skin and resolved lesions of psoriasis patients [111]. Furthermore, methylation differences between never-lesional psoriatic skin and healthy skin from volunteers, involving the Wnt and cadherin pathway genes, have also been identified and suggest that uninvolved skin might signify a pre-psoriatic condition with underlying disease susceptibility [112].

Changes in DNA methylation have also been detected in peripheral blood mononuclear cells (PBMCs) of individuals with psoriasis. In fact, DNMT1 is overexpressed in PBMCs from psoriatic patients, while the expression of methyl-CpG binding domain protein 2 (MBD2) and methyl-CpG binding protein 2 (MeCP2), important regulators of DNA methylation, is significantly decreased [113]. Furthermore, differential methylation of the promotors of genes involved in cell interaction, signal transduction, nucleotide degradation, skin differentiation, and cell motility have also been observed in mesenchymal stem cells of psoriasis patients [114].

DNA methylation can also play a role in the development of psoriatic arthritis: different DNA methylation patterns in CD8+ T cells may facilitate distinction of purely cutaneous psoriasis and psoriatic arthritis [115,116,117]. The published studies are few and the size samples small, so further research on this issue is needed.

#### 3.2.3. Histone Modification

Histones are greatly conserved proteins present in cell nuclei that participate in gene regulation. H2A, H2B, H3, and H4 are core histones, which are assembled into an octamer with 146 base pairs of DNA wrapped around it. H1 histones serve as linkers of DNA on the entry and exit sites of nucleosomes. Post-transcriptional modifications of histones can change the accessibility of DNA sequences and consequently alter their transcription [108]. Several histone modifications have been described in psoriasis skin with potential pathogenetic implications.

Histone methylation can lead to either an active or suppressed state of transcriptional activity depending on the methylation site and number of methyl groups introduced. Methylation of H3K9 can modify IL23 expression in keratinocytes [118], whereas H3K4 methylation has been detected in PBMCs from psoriasis patients [119]. Reduced levels of methylated H3K27, produced by the Jmjd3 demethylase, are associated with Th17 differentiation [120]. Moreover, methylation levels of H3K27 and H3K4 are different between responders and non-responders to biological treatment, suggesting that these changes could influence response to treatment [119].

Histone acetylation provokes weaker interactions between histones and DNA, resulting in open chromatin and facilitation of active transcription [4]. Reduced histone H4 acetylation in PBMCs from psoriasis patients compared to normal controls have been found [119].

As compared with healthy skin, lesional psoriasis skin exhibits dysregulated histone acetyltransferases (HATs) and histone deacetylases (HDACs), the enzymes that sustain the general equilibrium between histone acetylation and deacetylation [108]. In fact, HDAC-1 has been found to be upregulated in psoriatic skin, which could induce the over-expression of VEGF, proliferation of endothelial cells, and keratinocyte survival [121]. Elevated H3K9 and H3K27 acetylation in the *IL17A* promoter region in immune cells of psoriatic patients have been observed, leading to Th17 differentiation and psoriasis development [122].

The BET protein family controls the transcription of an extensive range of proinflammatory and immunoregulatory genes by recognizing acetylated histones and enlisting transcription factors, prompting transcription initiation and elongation [123]. Inhibition of BET proteins has been shown to decrease the expression of RORC, IL-17A and IL-22, all of them important psoriasis pro-inflammatory factors, representing a potential new therapy for psoriasis [124].

## 4. Conclusions

This narrative review highlights the complexity of the genetics and epigenetics of psoriasis, involving a vast number of genes coding for proteins implicated in the activity of the innate and adaptative immune systems along with the skin barrier. The rapid technological development of genetic studies has led to enhanced comprehension of the physiopathology of psoriasis and facilitated the emergence of new drugs. Since genetic variants are associated with disease susceptibility, they could be useful to predict disease risk [125]. Polygenic risk scores consist of weighted sums of the individual risk alleles that might identify those individuals at high risk of psoriasis development or predict its severity, association with psoriatic arthritis or even treatment outcomes. Even though polygenic risk scores have been used in research studies, their utility in clinical practice has yet to be established [125]. It is worth mentioning that, although genetics of generalized pustular psoriasis has been explored, the vast majority of the research is conducted in plaque psoriasis, the most common type of psoriasis. Thus, there is a need for studies assessing the genetics and epigenetics of other different phenotypes of psoriasis.

Epigenetic changes also provide insight into disease pathology and point towards future treatment approaches. Furthermore, since epigenetic factors contribute to the physiopathology of psoriasis, there is growing interest in the development of new drugs targeting epigenetic mechanisms, such as inhibition of DNA methylation [126], histone deacetylation [127], or modification of non-coding RNA [128]. Epigenetic markers could also serve as potential diagnostic or therapeutic response biomarkers. However, the multitude of distinct cell types implicated in psoriatic disease amplifies the complexity of unraveling the role of the epigenome; this review provides only a glimpse and further research is needed in this field.

## Figures and Tables

**Figure 1 genes-14-01619-f001:**
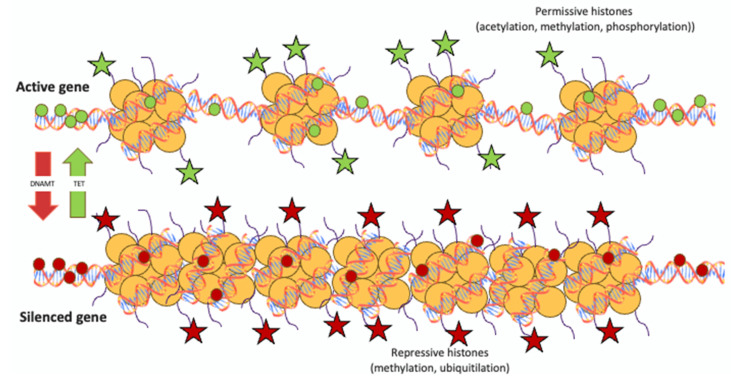
Epigenetic alterations that modify gene transcription: DNA methylation and histone modification. Euchromatin and heterochromatin. Unmethylated CpG islands (greencircles), permissive histone modifications (green stars) and loosechromatin structure promote gene transcription. Conversely, DNAmethylation (red circles), repressive histone modifications (red stars) and condensed structure prevent transcription. Although governed bydistinct enzymes, there is cooperativity and interaction between thedifferent epigenetic modifications. TET = Ten-eleven translocationdioxygenase; DNAMT = DNA methyltransferase.

**Table 1 genes-14-01619-t001:** Genes associated with psoriasis.

Pathway	Gene	Function
**Antigen presentation**	*HLA-C*0602*	Antigen presentation
	*ERAP1*	Modification of MHC-I-binding peptides
**Th1 Signaling Pathway**	*IL12B*	p40 subunit of IL12
	*TYK2*	Downstream molecule of IL12 receptor
	*ZC3H12C*	Macrophage activation
	*STAT5A/B*	Signaling pathway of IL2 familiy cytokines
	*ILF3*	IL2 expression in T-cells
**Th17 Signaling Pathway**	*TYK2*	Downstream molecule of IL23 receptor
	*JAK2*	Downstream molecule of IL23 receptor
	*STAT3*	Downstream molecule of IL23 receptor
	*SOCS1*	Th17 differentiation
	*ETS1*	Th17 differentiation
	*IL17RD*	IL17 receptor
	*IL22*	Differentiation and proliferation of keratinocytes
	*TRAF3IP2*	Signaling pathway of IL17A/F
	*KLF4*	Regulation of IL17A production
**Innate immunity**	*C-REL*	NF-kB pathway activation
	*TRAF3IP2*	NF-kB pathway activation
	*CARD14*	NF-kB pathway activation
	*MICA*	NK, NKT and T-cells activation
	*TNFAIP3*	NF-kB pathway downregulation
	*TNIP1*	NF-kB pathway downregulation
	*NFKBIA*	NF-kB pathway downregulation
	*DDX58*	INF pathway and antiviral response
	*IFIHI*	INF pathway and antiviral response
**Skin barrier function**	*DEFB4*	Secretion of β-defensins
	*LCE3B/C*	Epidermis differentiation and hyperproliferation
	*GJB2*	Connexin 26, epidermal gap junction

**Table 2 genes-14-01619-t002:** Main non-coding RNAs associated with psoriasis.

Epigenetic Change		Function
**miRNA downregulation**	miRNA125b	Keratinocyte proliferation and differentiation
	miR-203	Keratinocyte proliferation
	mir-383	Keratinocyte apoptosis and inflammation
	214-3p	Cell cycle check-points and keratinocyte proliferation
	miR-125a-5p	Keratinocyte proliferation
miRNA upregulation	miR-378a	Psoriatic inflammation
	Mir-31	Keratinocyte proliferation
	mir-210	Inflammation
	miR-200c	Associated with PASI
	miR-155	Psoriatic inflammation
lncRNA upregulation	lncRNA-RP6- 65G23.1	Immune response, keratinocyte proliferation, apoptosis suppression
	MIR31HG	Keratinocyte proliferation
	MSX2P1	Keratinocyte proliferation
	XIST	Keratinocyte proliferation
	FABP5P3	Keratinocyte proliferation and inflammation
	KLDHC7B-DT	Keratinocyte proliferation and inflammation
	SPRR2C	Keratinocyte proliferation and apoptosis
lncRNA downregulation	MEG3	Keratinocyte proliferation and apoptosis
	GAS5	Related to psoriasis severity
	PRINS	Keratinocyte proliferation and inflammation
	NEAT1	Keratinocyte proliferation
circRNA downregulation	circRAB3B	Keratinocyte proliferation
circRNA upregulation	circOAS3	Keratinocyte proliferation and apoptosis
	circEIF5	Keratinocyte proliferation
	circ_0060531	Keratinocyte proliferation, migration &inflammation
	hsa_circ_0003738	Treg modulation
	hsa_circ_0061012	Keratinocyte proliferation and migration

## Data Availability

Not applicable.

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
