# Peer review of "Genetic and Epigenetic Mechanisms of Psoriasis"

_genes, 2023, doi:10.3390/genes14081619_

Round 1

Reviewer 1 Report

A well written and comprehensive update on Psoriasis genetics. One would like to rearrange the flow a bit and Suggest to present non-coding RNAs before the Epigenetics section which  could be numbered as 4 followed by DNA methylation and Histone modifications which should be contained within the Epigenetics section as 4.1 and 4.2 . One could also present the no-coding RNAs as the first part of the epigenetic section as 4.1 followed by DNA methylation and Histone modifications.

Conclusions would then be 5. Maybe elaborate a bit more on Future Perspectives highlighting the unmet need for more detailed genetics on the different phenotypes of psoriasis. Now only pustular psoriasis is separated ( understandably since this is where there is some data ) but underline that so far most genetics data derive from moderate to severe plaque psoriasis and we do not know much of genetic factors that may modify the disease course in a positive way 

Author Response

Dear reviewers and editorial team,

Thank you for considering our review for publication. We are grateful to the reviewers for pointing out some issues regarding this manuscript. The point-by-point explanation of what has been changed in response to the referees’ concerns is provided below. Changes translated into the text appear into the text highlighted in yellow in the revised version of the manuscript.

Moreover, in the initial version of the manuscript, we submitted a figure for consideration by the reviewers. We are reattaching the figure again in case you deem it relevant to include in the review.

We hope that all these clarifications fulfil the requirements of the referees and editors thus making the manuscript acceptable for publication.

Reviewer 1

A well written and comprehensive update on Psoriasis genetics.

One would like to rearrange the flow a bit and Suggest to present non-coding RNAs before the Epigenetics section which  could be numbered as 4 followed by DNA methylation and Histone modifications which should be contained within the Epigenetics section as 4.1 and 4.2 . One could also present the no-coding RNAs as the first part of the epigenetic section as 4.1 followed by DNA methylation and Histone modifications. Conclusions would then be 5.

We completely agree with the reviewer that sections numbers appear confusing in the current version of the manuscript. We have modified them accordingly and, following the reviewer’s suggestion, moved the “non-coding RNAs” section as the first part of the epigenetic section.

Maybe elaborate a bit more on Future Perspectives highlighting the unmet need for more detailed genetics on the different phenotypes of psoriasis. Now only pustular psoriasis is separated ( understandably since this is where there is some data ) but underline that so far most genetics data derive from moderate to severe plaque psoriasis and we do not know much of genetic factors that may modify the disease course in a positive way.

We thank the reviewer for pointing out this issue. According to the recommendation, we have added a comment highlighting the need for more research in other subtypes of psoriasis.

Reviewer 2

The review: Genetic and epigenetic mechanisms of psoriasis consists of 4 main chapters, of which the most extensive part of Results is additionally divided into 12 subchapters. This procedure made the information provided in the article orderly.

Introduction. It clearly defines the problem of psoriasis, although the data contained in the introduction come from 2015. I think the figures may have changed a bit since this year. Can the authors provide the latest statistics on the disease? e.g. from Global burden of disease. In addition, I suggest you read the publications that indicate factors that increase the risk of psoriasis:

  1. Santus, P.; Rizzi, M.; Radovanovic, D.; Airoldi, A.; Cristiano, A.; Conic, R.; Petrou, S.; Pigatto, P.D.; Bragazzi, N.C.; Colombo, D.; et al. Psoriasis and respiratory comorbidities: The added value of fraction of exhaled nitric oxide as a new method to detect, evaluate, and monitor psoriatic systemic involvement and therapeutic efficacy. BioMed Res. Int. 2018, 23, 3140682.
  1. Conic, R.; Damian, G.; Schrom, K.P.; Ramser, A.E.; Zheng, C.; Xu, R; McCormick, T.S.; Cooper, K.D. Psoriasis and psoriatic arthritis cardiovascular disease endotypes identified by red blood cell distribution width and mean platelet volume. J. Clin. Med. 2020, 9, 186.

We agree with the reviewer that data should be updated. We have expanded the introduction section accordingly with more recent information, as well as included the suggested publications.

Results. As the authors mention several times, the process of psoriasis pathogenesis is extremely complicated and still not fully understood, which still prevents the creation of a single effective therapy. In addition, there are still new reports indicating completely different, new (different in other types of disease) factors or cells that can regulate the development of psoriasis. And although the most well-known regulatory axis of TNF/IL-23/IL-17 has been clearly characterized by the authors, the elements regulating its functioning are, among others, neutrophils and their traps (NETs). The authors mention them indirectly (MPO gene) at the GPP (which, apart from the presented article, is also confirmed by the work: 10.1111/1346-8138.16700). Do the authors know about the role of genes encoding NET proteins in the pathogenesis of psoriasis? Haven't the expressions of genes encoding NET proteins already been investigated within this phenomenon?

We thank the reviewer for this comment that enriches the quality of the manuscript. As stated by the reviewer, the role of neutrophil extracellular traps in the pathogenesis of psoriasis has already been investigated. The formation of NETs has been found to be increased in psoriatic skin. Specifically, the formation of NET is heightened in cases of loss of function of the IL36 gene, and NETs have been linked to the activation of the cutaneous inflammatory response via epidermal TLR4/IL-36R crosstalk. Furthermore, it has been observed that NET promotes the expression of defensins in the keratinocytes of patients with psoriasis. While it appears that NETs induce cytokine gene expression that favors Th17 differentiation and IL17A expression in patients with mutations in the TRAF3IP2 gene (included in the revised version of the manuscript), as far as we know, possible variations in NET-related genes related with the risk of psoriasis have not been strictly studied.

References. References include 120 items, including more than 30% from the last 3 years, which proves the authors' excellent knowledge of the problem.

In my opinion, the article is well thought out, carefully planned and perfectly executed.

We are grateful to the reviewer for this comment.

Reviewer 3

We congratulate the authors for their very thorough analysis of the existing literature on the genetic and epigenetic mechanisms of psoriasis. For me, as a researcher, this manuscript is a very good compendium of current knowledge, which I will undoubtedly quote.

I read the manuscript with a great pleasure.

The structure of the manuscript is correct, the content is understandable to the reader. The authors draw the right conclusions from their research work. I have no comments.

We are grateful to the reviewer for this comment.

Reviewer 2 Report

The review: Genetic and epigenetic mechanisms of psoriasis consists of 4 main chapters, of which the most extensive part of Results is additionally divided into 12 subchapters. This procedure made the information provided in the article orderly.

Introduction. It clearly defines the problem of psoriasis, although the data contained in the introduction come from 2015. I think the figures may have changed a bit since this year. Can the authors provide the latest statistics on the disease? e.g. from Global burden of disease.

In addition, I suggest you read the publications that indicate factors that increase the risk of psoriasis:

1. Santus, P.; Rizzi, M.; Radovanovic, D.; Airoldi, A.; Cristiano, A.; Conic, R.; Petrou, S.; Pigatto, P.D.; Bragazzi, N.C.; Colombo, D.; et al. Psoriasis and respiratory comorbidities: The added value of fraction of exhaled nitric oxide as a new method to detect, evaluate, and monitor psoriatic systemic involvement and therapeutic efficacy. BioMed Res. Int. 2018, 23, 3140682.

2. Conic, R.; Damian, G.; Schrom, K.P.; Ramser, A.E.; Zheng, C.; Xu, R; McCormick, T.S.; Cooper, K.D. Psoriasis and psoriatic arthritis cardiovascular disease endotypes identified by red blood cell distribution width and mean platelet volume. J. Clin. Med. 2020, 9, 186.

Results. As the authors mention several times, the process of psoriasis pathogenesis is extremely complicated and still not fully understood, which still prevents the creation of a single effective therapy. In addition, there are still new reports indicating completely different, new (different in other types of disease) factors or cells that can regulate the development of psoriasis. And although the most well-known regulatory axis of TNF/IL-23/IL-17 has been clearly characterized by the authors, the elements regulating its functioning are, among others, neutrophils and their traps (NETs). The authors mention them indirectly (MPO gene) at the GPP (which, apart from the presented article, is also confirmed by the work: 10.1111/1346-8138.16700). Do the authors know about the role of genes encoding NET proteins in the pathogenesis of psoriasis? Haven't the expressions of genes encoding NET proteins already been investigated within this phenomenon?

References. References include 120 items, including more than 30% from the last 3 years, which proves the authors' excellent knowledge of the problem.

In my opinion, the article is well thought out, carefully planned and perfectly executed.

Author Response

(The authors gave the same response as above.)

Reviewer 3 Report

We congratulate the authors for their very thorough analysis of the existing literature on the genetic and epigenetic mechanisms of psoriasis. For me, as a researcher, this manuscript is a very good compendium of current knowledge, which I will undoubtedly quote.

I read the manuscript with a great pleasure.

The structure of the manuscript is correct, the content is understandable to the reader. The authors draw the right conclusions from their research work. I have no comments.

Author Response

(The authors gave the same response as above.)
